# ADCY5 Gene Affects Seasonal Reproduction in Dairy Goats by Regulating Ovarian Granulosa Cells Steroid Hormone Synthesis

**DOI:** 10.3390/ijms26041622

**Published:** 2025-02-14

**Authors:** Chenbo Shi, Fuhong Zhang, Qiuya He, Jianjun Man, Yuanpan Mu, Jianqing Zhao, Lu Zhu, Juan J. Loor, Jun Luo

**Affiliations:** 1Shaanxi Key Laboratory of Molecular Biology for Agriculture, College of Animal Science and Technology, Northwest A&F University, Yangling 712100, China; shichenbo@nwafu.edu.cn (C.S.); zhang_fuhong@nwafu.edu.cn (F.Z.); heqiuya@nwafu.edu.cn (Q.H.); manjianjun@nwafu.edu.cn (J.M.); 2022055443@nwafu.edu.cn (Y.M.); zhao2021@nwafu.edu.cn (J.Z.); zhulu@nwafu.edu.cn (L.Z.); 2Mammalian NutriPhysioGenomics, Department of Animal Sciences and Division of Nutritional Sciences, University of Illinois, Urbana, IL 61801, USA; jloor@illinois.edu

**Keywords:** dairy goats, seasonal reproduction, *ADCY5*, RNA-seq, ovarian granulosa cells

## Abstract

Follicle development in dairy goats is lower after induced estrus during the non-breeding season, reducing conception rates and challenging year-round milk supply. This study investigated follicle development during the breeding and non-breeding seasons and explored molecular mechanisms for variations in the proportions of follicles of different sizes using ovarian RNA-seq and in vitro experiments. Induced estrus during the non-breeding season used a simulated breeding season short photoperiod and male effect methods, while the male effect method was used during the breeding season. This study identified an increase in follicle size during the breeding season and performed RNA-seq on ovaries to explore the underlying causes. The RNA-seq analysis elucidated pathways associated with cellular and hormonal metabolism and identified adenylyl cyclase 5 (*ADCY5*) as a key differentially expressed gene. In vitro experiments demonstrated that interfering with *ADCY5* in ovarian granulosa cells (GCs) reduced steroid synthesis. Conversely, the overexpression of *ADCY5* increased steroid synthesis. *ADCY5* affects the biological function of GCs and consequently influences follicle development through the cAMP-response element binding protein (CREB) and p38 mitogen-activated protein kinase phosphorylation (MAPK) pathways. Overall, our findings demonstrate that follicle development in dairy goats differs between the breeding and non-breeding seasons and that the differential expression levels of the *ADCY5* gene contribute to this discrepancy.

## 1. Introduction

In the northwestern region of China, located in the Northern Hemisphere, the anestrous period for goats spans from January to June. During this time, the goats experience a prolonged photoperiod and enter a reproductive quiescence, commonly termed the non-breeding season. Conversely, from July to December, the photoperiod shortens, and the goats transition into a reproductive phase known as the breeding season. The periodic anestrus of dairy goats poses a challenge to guaranteeing milk’s long-term availability and satisfying the year-round market demands. Inducing goats to reach reproductive status in the non-breeding season can ensure a year-round milk supply. Inducing estrus in dairy goats holds considerable importance, particularly during the non-breeding season. Various estrus induction protocols are widely used in production, including the use of progesterone sponges in conjunction with prostaglandin F2α (PGF2α), which have proven effective in inducing estrus [1]. Additionally, the combination of melatonin (MEL) and progesterone (P4) has been shown to successfully induce estrus and ovulation, thereby enhancing reproductive success rates [2]. Beyond exogenous hormone treatments, behavioral stimuli are also employed to induce estrus. For instance, the introduction of sexually active male goats to anestrous females during the non-breeding season can effectively stimulate estrus and ovulation through male-female interactions, a phenomenon referred to as the “male effect” [3]. As short-day breeders, dairy goats’ reproductive activities are influenced by photoperiods. Introducing a short photoperiod is another method for inducing estrus; for example, exposing ewes to artificial long daylighting followed by a natural photoperiod can effectively induce estrus in ewes during the non-breeding season [4].

However, significant variability in reproductive performance between breeding and non-breeding seasons is widely observed, with lower ovulation and pregnancy rates in ewes following induced estrus during the non-breeding season, likely attributable to the seasonal effects of reproduction on follicle development and embryo quality [5,6]. Estrus induction protocols often result in higher pregnancy rates during the breeding season than the non-breeding season; even common schemes like the P4 scheme, P4+ pregnant mare serum gonadotropin (PMSG) scheme, and PGF2α scheme still lead to lower pregnancy rates during the non-breeding season [5,6]. It has been reported that after the end of the breeding season in sheep, even with follicle-stimulating hormone (FSH) stimulation, the proportion of small follicles to total follicles increases [7,8,9,10]. Seasonal changes also affect follicle function and embryo viability, as evidenced by an increased fertilization failure rate and the lower quality and viability of embryos produced during the late breeding season in April [11]. Therefore, investigating the biological processes associated with oocyte maturation and ovulation across various seasons, along with the seasonal variations in follicle development, is highly significant.

In this study, during the non-breeding season, the dairy goats were exposed to a simulated short-day photoperiod similar to that of the breeding season, while also receiving the male effect protocol to induce estrus. During the breeding season, we utilized the male effect protocol to induce estrus in the dairy goats. We hypothesize differences in follicular development in dairy goats between the breeding and non-breeding seasons, and this study aims to investigate these differences. The subsequent objective is to elucidate the potential causes of these differences through transcriptome sequencing of ovarian tissue and in vitro cellular experiments. This study aims to provide new insights into the seasonal reproduction differences of dairy goats. The ultimate aim is to enhance the reproductive performance of dairy goats and facilitate continuous milk production throughout the year.

## 2. Results

### 2.1. Follicle Diameter and Follicle Number

In Experiment 1 (non-breeding season), there were 20 dairy goats under natural long daylight treatment (control group, C1 group) and 20 dairy goats that received simulated short daylight mimicking the breeding season along with male effect treatment (LM1 group). In Experiment 2 (breeding season), there were 20 dairy goats under natural short daylight treatment (control group, C2 group) and 20 dairy goats receiving both natural short daylight and male effect treatment (M2 group). An analysis was conducted on the follicle count and size of the ovaries, as shown in Table 1. Dairy goats exposed to natural short daylight during the breeding season had a significantly higher number of large-sized follicles compared to those exposed to natural long daylight during the non-breeding season (*p* < 0.05). Conversely, the number of small-sized follicles in dairy goats exposed to natural short daylight during the breeding season was significantly lower than in those exposed to natural long daylight during the non-breeding season (*p* < 0.05). Similarly, the number of large-sized follicles and the total follicle count in dairy goats that underwent induced treatment during the breeding season were significantly higher than those in dairy goats that underwent induced treatment during the non-breeding season (*p* < 0.05). In conclusion, although the non-breeding season simulated the inducing factors of the breeding season, dairy goats during the breeding season had more large follicles compared to those in the non-breeding season. Therefore, it can be inferred that the follicle development in estrous dairy goats is superior to that in non-breeding season goats.

### 2.2. Transcriptome Analysis of Goat Ovarian Tissue

To investigate the molecular mechanisms influencing follicle development, we performed transcriptome sequencing of the ovaries. The RNA-seq analysis uncovered 3542 up-regulated and 2944 down-regulated DEGs in LM1 vs. M2. These DEGs include steroidogenic acute regulatory protein (*StAR*, |log2FC| = 1.93), 3 beta-hydroxysteroid dehydrogenase (*3βHSD*, |log2FC| = 3.65), Cytochrome P450 family 11 subfamily A member 1 (*CYP11A1*, |log2FC| = 2.54), epidermal growth factor receptor (*EGFR*, |log2FC| = 1.21), adenylyl cyclase 5 (*ADCY5,* |log2FC| = 2.06), AKT serine/threonine kinase 3 (*AKT3*, |log2FC| = 3.21), and mitogen-activated protein kinase 12 (*MAPK12*, |log2FC| = 1.14) (Figure 1A and Appendix A). In C1 vs. C2, there were 930 up-regulated and 608 down-regulated DEGs. These DEGs include *StAR* (|log2FC| = 1.01), *ADCY5* (|log2FC| = 0.59), bone morphotic protein 15 (*BMP15*, |log2FC| = 0.89), follicle-stimulating hormone receptor (*FSHR*, |log2FC| = 3.37), and Cytochrome P450 17A1 (*CYP17A1*,|log2FC| = 3.17) (Figure 1B and Appendix A). There are 569 shared DEGs, including low-density lipoprotein receptor (*LDLR*), phosphoinositide-3-kinase regulatory subunit 1 (*PIK3R1*), *ADCY5*, growth differentiation factor 10 (*GDF10*), hydroxysteroid dehydrogenase 17 beta 1 (*HSD17B1*), and insulin-like growth factor binding protein 3 (*IGFBP3*) (Figure 1C).

We performed pathway enrichment analysis in the KEGG database to characterize the shared DEGs (Figure 1D). We found enriched pathways including metabolic pathways, steroid biosynthesis, GnRH secretion, ovarian steroidogenesis, cortisol synthesis and secretion, camp signaling pathway, as well as neuro-related GABAergic synapse and serotonergic synapse. The findings suggest that the reproductive performance differences between the breeding and non-breeding seasons may be related to ovarian hormone synthesis, metabolism, and neurology. Therefore, subsequent research will focus on the DEGs involved in hormone synthesis, metabolism, and neurology.

### 2.3. ADCY5 Is Involved in Regulating the Proliferation of GCs

In our RNA-seq results, we found that *ADCY5* as a DEG showed a higher expression level during the breeding season. Considering the crucial role of GCs in follicle development, oocyte maturation, and sex hormone secretion, we further investigated the effects of *ADCY5* on GC proliferation and steroid synthesis metabolism. The CD sequence of *ADCY5* was sequenced and identified, spanning 3780 bp in the goat *ADCY5*-CDs region and encoding a protein with 1259 amino acids (Figure 2A and Appendix A). RNA-FISH analysis results also demonstrated the expression of *ADCY5* in GCs (Figure 2B).

When transfecting siRNA into GCs, there was a significant decrease in *ADCY5* mRNA expression (*p* < 0.05) and protein level (*p* < 0.01) (Figure 3A,B). qPCR and Western blot were utilized to analyze the mRNA and protein expression levels of the cell cycle-related genes *CDK1* and *CDK2*. The findings revealed a significant decrease in the mRNA levels of *CDK1* (*p* < 0.001) (Figure 3C), as well as in the protein expression of CDK1 (*p* < 0.01) (Figure 3D) following the interference of *ADCY5*. The CCK8 analysis results revealed that interfering with the expression of *ADCY5* would lead to a decrease in the proliferation activity of GCs (*p* < 0.01) (Figure 3E).

Following the transfection of GCs with the *ADCY5* overexpression vector, there was a significant increase in both the mRNA (*p* < 0.001) and protein expression levels (*p* < 0.05) of *ADCY5* (Figure 4A,B). qPCR analysis revealed that *ADCY5* overexpression markedly elevated the mRNA expression levels of *CDK1* (*p* < 0.001) and *CDK2* (*p* < 0.05) in GCs (Figure 4C). Additionally, Western blot analysis demonstrated a significant increase in the protein expression level of CDK1 (*p* < 0.05) (Figure 4D) upon *ADCY5* overexpression. Furthermore, CCK8 assays indicated that *ADCY5* overexpression significantly enhanced the proliferation activity of GCs (*p* < 0.001) (Figure 4E).

Comparable outcomes were achieved utilizing EDU detection. The suppression of *ADCY5* led to a statistically significant decrease in the number of positive cells (*p* < 0.05) (Figure 5A,C), whereas its overexpression resulted in a statistically significant increase in the number of positive cells (*p* < 0.01) (Figure 5B,D).

### 2.4. ADCY5 Is Involved in Regulating the Steroidogenesis of GCs

Subsequently, the involvement of *ADCY5* in ovarian hormone synthesis was examined. Interference with *ADCY5* significantly reduced the concentrations of estradiol (E2) (*p* < 0.05) and P4 (*p* < 0.05) in the cell culture media (Figure 6A,B), while overexpression significantly increased the concentrations of E2 (*p* < 0.01) and P4 (*p* < 0.05) (Figure 6E,F). Mechanistically, the downregulation of *ADCY5* significantly decreased the mRNA and protein expression of key steroid synthesis genes (Figure 6C,D), including *StAR*, *CYP11A1*, Cytochrome P450 19A1 (*CYP19A1*), and *3βHSD*, while its overexpression had the opposite effect (Figure 6G,H).

### 2.5. ADCY5 Is Responsible for the Activation of the CREB

The Western blot results showed that down-regulated *ADCY5* significantly reduced the phosphorylation level of CREB (*p* < 0.01) (Figure 7A,B), while overexpressing *ADCY5* significantly increased the phosphorylation level of CREB (*p* < 0.01) (Figure 7C,D).

Furthermore, since ADCY5 can promote the synthesis of cAMP, we employed the PKA activator 8-Br-cAMP, a cell-permeable cAMP analog, to compare its effects with those observed from *ADCY5* overexpression. The results showed that 8-Br-cAMP significantly promoted the phosphorylation of CREB (Appendix A). Interestingly, 8-Br-cAMP also promotes cell proliferation (Supplement Appendix A). We investigated the impact of ADCY5 on the expression and phosphorylation levels of p38, given that CREB phosphorylation can be activated by MAPK p38. The interference of *ADCY5* significantly reduced p-p38 levels (*p* < 0.01) (Figure 7A,B), while the overexpression of *ADCY5* markedly increased p-p38 levels (*p* < 0.01) (Figure 7C,D). The presence of 8-Br-cAMP stimulated the phosphorylation of p38 (Appendix A). The above results indicate that ADCY5 impacts the biological processes of GCs through the PKA/CREB and p38/CREB pathways.

## 3. Discussion

Our study observed that the follicle development in dairy goats induced during the non-breeding season is inferior to that in the breeding season, primarily reflected in the size of the follicles. This may be a reason for the lower pregnancy rates in dairy goats during the non-breeding season in livestock production. Studies have shown that the follicle size of dromedary camels (*Camelus dromedarius*) during the breeding peak season is larger, with over 50% of female camels having ovarian follicles not exceeding 10 mm during the non-breeding season [12]. The follicle size and number of Boer goats (*Capra hircus*) during the non-breeding season are less compared to the breeding season [13]. The slow development of these non-reproductive stage follicles may be due to the overall lower LH levels during the non-reproductive stage [14]. We hypothesize that the decreased follicular development is responsible for the decreased pregnancy rate during non-breeding seasons. Nonetheless, the regulation of follicular development in non-breeding seasons is intricate, necessitating further exploration of potential underlying mechanisms. The size of ovarian follicles is intrinsically linked to the developmental potential of oocytes, with its regulation being governed by a multitude of molecular mechanisms, such as the regulation of gene expression, signaling pathways, and components of follicular fluid. Furthermore, throughout follicular development, the gene expression activities of granulosa cells play a crucial role in sustaining the intricate network of signaling pathways within the follicles.

In our study, we conducted ovarian transcriptome analysis to explore the potential molecular mechanisms underlying the differences in follicle development between the breeding and non-breeding seasons. Our findings revealed associations of various metabolic and hormone-related pathways with female reproduction, in addition to enrichments in neural-related pathways. The metabolic pathways have been confirmed to be associated with oocyte division [15], and metabolic disorders have been widely reported in the study of polycystic ovary syndrome in humans [16,17]. The steroid biosynthesis in the ovary is crucial for ovarian function [18], and granulosa cells influence the process of follicular development by producing steroids [19]. Progesterone-mediated oocyte maturation has also been shown to play a role in sheep follicle maturation and the in vitro meiotic maturation of porcine oocytes [20,21]. Based on the findings of the KEGG enrichment analysis, we inferred that while estrus was induced in non-breeding seasons, seasonality still affects follicle development, consequently affecting both pregnancy rates and reproductive capacity. This may be associated with ovarian metabolism, hormone synthesis, and neural system functionality.

The RNA-seq analysis identified 569 shared DEGs, and upon reviewing the studies of these DEGs, we found that *ADCY5* may be associated with the seasonality of animal reproduction [22]. Oocyte maturation is regulated by intracellular cAMP levels through the activity of endogenous adenylyl cyclase, suggesting that ADCY may be involved in the early stages of oocyte meiosis [23]. The whole genome association study (GWAS) identified *ADCY5* as a potential gene related to bovine fertility, and its genetic polymorphism affects ovarian width and the diameter of the corpus luteum [24]. *ADCY5* may also play a significant role in regulating seasonal reproduction in sheep [22]. Genes responsible for regulating ovarian steroids are crucial for the maintenance of follicular development and luteal function. The quality of mature follicles in the ovary affects the reproductive performance of cows [25]. GCs proliferate vigorously in primary and secondary follicles [26], and their proliferation is crucial for the normal development of oocytes [27]. Our research has found that *ADCY5* affects the proliferative activity of GCs as well as the expression of the cell cycle proteins CDK1 and CDK2. CDK1 and CDK2 are the major regulators of the cell cycle [28]. CDK1 is essential for mammalian cell proliferation [28], and conditional knockout of CDK1 in mice leads to proliferation defects [29]. CDK2 also plays a role in cell proliferation; miRNA-125 inhibits the proliferation of cochlear progenitor cells by negatively targeting CDK2 [30], and inhibiting CDK2 suppresses the proliferative activity of podocytes [31]. StAR facilitates the transfer of cholesterol from the outer mitochondrial membrane to the inner membrane, catalyzing the synthesis of pregnenolone by CYP11A1 and the synthesis of androgens through membrane-bound CYP17A1 [32]. Subsequently, upon diffusion to granulosa cells (GCs), CYP19A1 catalyzes the formation of E2 [33]. In this study, the secretion of P4 and E2 by GCs is modulated by *ADCY5*. P4 priming during follicular development has been demonstrated to enhance oocyte competence, resulting in increased rates of cleavage and embryo development in sheep [34]. Additionally, another investigation examined the impact of estradiol on bovine cumulus–oocyte complexes, revealing that E2 supplementation is essential for preserving oocyte quality and maturity [35]. In this study, we have inferred that *ADCY5* may modulate the development of ovarian follicles by impacting the proliferation of granulosa cells and potentially by controlling the synthesis of ovarian steroids.

Our research shows that ADCY5 regulates the activity of CREB and p38. ADCY5 activates cAMP-dependent protein kinase A (PKA) by promoting the synthesis of cAMP. The activation of PKA induces the expression of StAR [36], thereby stimulating acute steroid synthesis, as PKA enhances the phosphorylation of CREB [37]. The activation of CREB promotes cellular steroid synthesis and proliferation [38,39,40], while its dephosphorylation induces cell apoptosis [26]. CREB, as one of the main downstream transcription factors, is also regulated by p38 MAPK, as the phosphorylation of p38 depends on the activation of PKA [41,42]. Our research reveals that ADCY5 modulates the activity of CREB and p38, possibly by regulating cAMP levels. The PKA agonist 8-Br-cAMP to enhance the phosphorylation of CREB and p38 further substantiates this assertion. Consequently, we posit that ADCY5 regulates CREB activity through the cAMP/PKA and p38 MAPK pathways, influencing the synthesis of steroid hormones in GCs.

In conclusion, our study provides evidence that, despite inducing estrus by simulating the light conditions of the breeding season during the non-breeding season, dairy goats exhibit larger follicle sizes in the breeding season. Through ovarian transcriptome sequencing and in vitro experiments, it was revealed that the *ADCY5* gene regulates the biological functions of GCs through the PKA/CREB and p38 MAPK pathways. Consequently, we propose that ADCY5 modulates CREB activity via the cAMP/PKA and p38 MAPK signaling pathways, thereby affecting steroid hormone synthesis in GCs and facilitating follicular development during the breeding season. This mechanism may elucidate the observed increase in pregnancy rates among dairy goats during this period. Our research identifies a potential target for enhancing reproductive performance in dairy goats during the non-breeding season.

## 4. Materials and Methods

### 4.1. Animals and Diets

This study was conducted in Guanzhong region, Shaanxi, China (34°16′ N, 108°4′ E). The region has distinct four seasons, characterized by a continental monsoon climate, with an average annual temperature of 12 °C. Following the summer solstice, the duration of daylight diminishes, prompting the dairy goats in the region to enter the breeding season. Conversely, post the winter solstice, the daylight duration increases, leading the dairy goats in the area to transition into the non-breeding season. The peak breeding season for dairy goats in the region is from August (14 h of daylight) to October (12 h of daylight), and they clearly exhibit anestrus status from March (12 h of daylight) to May (14 h of daylight).

The goat breed is Xinnong Saanen dairy goats; all females were multiparous and ranged in age from 2.5 to 3.5 years. Average body weight and average feed intake of goats as shown in Table 2. The diet consists of corn silage and alfalfa hay for roughage. The concentrate includes soybean meal, corn grain, bran, trace mineral salt, and vitamin–mineral premix. The concentrate-to-roughage ratio is 4:6, and they have ad libitum access to water.

### 4.2. Experimental Design

Experiment 1 and Experiment 2 were conducted during the non-breeding season (March to May) and breeding season (August to October), respectively, with each experiment lasting 6 weeks.

In Experiment 1, there were 20 dairy goats under natural long daylight treatment (control group, C1 group) and 20 dairy goats that received simulated short daylight mimicking the breeding season along with male effect treatment (LM1 group). In Experiment 2, there were 20 dairy goats under natural short daylight treatment (control group, C2 group) and 20 dairy goats receiving both natural short daylight and male effect treatment (M2 group). Detailed information regarding the experimental seasons, grouping, number of dairy goats, body weight, and average feed intake can be found in Table 2.

### 4.3. Short Daylight Control and Male Effect

The short daylight control entailed a reduction of 15 min of light per week during experiment 1. This reduction in light duration was equivalent to the weekly light reduction during the experimental period in the breeding season. Shading curtains were installed on all doors and windows to regulate the lighting duration.

The treatment of male effect was that two male goats were placed beside the female goats’ pens and could reach the female through the fence but could not mate. Replaced two males every 3 days. Males were 2–3 years old and vigorous and had strong sexual desire. The staff used a tether to bring the male goats into the female goat pens every morning at 8 a.m. They were allowed to interact and court each other but could not mate, and this process lasted for 30 min.

### 4.4. Ovaries Collection and Follicle Counting

After observing estrous behavior, the ovaries were harvested under general anesthesia with xylazine hydrochloride (we administered a muscular injection at a dose of 2.5 µL/kg in accordance with the instructions; Huamu Animal Health Products Corporation, Changchun, China), and the ovaries were collected in a sterile operating room (C1: *n* = 3, LM1: *n* = 4, C2: *n* = 3, M2: *n* = 3). Following the surgery, the goats were placed in quiet and clean pens for 7 days, with daily iodine disinfection of the wounds. The follicle count was recorded, and the diameter of each follicle was measured using a ruler. Follicles with a diameter smaller than 2.5 mm and larger than 2.5 mm were categorized as small-sized and large-sized follicles, respectively. The ovaries were stored in liquid nitrogen for RNA sequencing and bioinformatics analysis.

### 4.5. RNA Extraction, Sequencing Libraries, and RNA-seq Analyses

Total RNA was extracted using Trizol reagent (Invitrogen, Carlsbad, CA, USA), and then the purity and concentration of the extracted RNA were assessed. According to the manufacturer’s manual, cDNA libraries and sequencing were performed at Allwegene Company (Beijing, China). Briefly, mRNAs were extracted from total RNA using oligo (dT) magnetic beads and then broken into short fragments. The fragmented mRNAs were reverse-transcribed into cDNAs, which were then amplified by PCR. The cDNA libraries were sequenced using Illumina HiSeq 2000 (Illumina Inc., San Diego, CA, USA). Clean data were obtained by removing reads containing adapters and low-quality reads from the raw data. Subsequent analyses were performed using the clean data. The clean reads were mapped to the *Capra hircus* reference genome Capra_hircus. ARS1 (https://ftp.ensembl.org/pub/release-111/fasta/capra_hircus) (accessed 20 December 2021). We calculated the number of clean reads of each mapped gene and normalized gene expression by RPKM method (reads per kilobase per million mapped reads). The reads per kilo bases per million reads (RPKM) method was adopted to evaluate the gene expression levels and genes with *p* < 0.05 and FoldChange > 1.5 were identified as DEGs [43]. R software (version 4.1.1) (Ggplot2 R package) was used for plot, and Kyoto Encyclopedia of Gene and Genome (KEGG) pathway analysis was performed using KOBAS 3.0 software. *p* < 0.05 is considered to be significantly enriched in DEGs. KEGG pathways with *p* < 0.05 were considered significantly enriched in DEGs.

### 4.6. GCs Isolation and Culture

Goat ovaries were collected from the local commercial slaughterhouse (the goats were slaughtered by neck bleeding in accordance with the prevailing national regulations pertaining to slaughterhouses). The collected ovaries will be rapidly placed in physiological saline solution containing 100 IU/mL of penicillin and 100 mg/mL of streptomycin (Takara, Beijing, China) and brought back to the laboratory within 1 h [44]. The follicular fluid was aspirated from healthy follicles with a diameter exceeding 1.5 mm using a 1 mL syringe. The collected fluid was centrifuged (1500× *g*, 3 min) to remove the supernatant and then added to cell culture medium (Gibco’s DMEM/F12 medium with 10% fetal bovine serum, 100 IU/mL of penicillin and 100 μg/mL of streptomycin) (Gibco, part of Thermo Fisher Scientific, Waltham, MA, USA) for cell suspension. The suspended cells were placed in a culture dish and cultured in a CO_2_ cell incubator (5% CO_2_ at 37 °C) [45]. We replaced the cell culture medium after 48 h in order to eliminate non-adherent cells. The cells from the ovaries of 5 goats were suspended and cultured in a culture dish. The cells used for experimental replicates were all from different goat ovaries.

### 4.7. RNA Interference and Overexpression of ADCY5

GCs were seeded into 6-well plates or 96-well plates for siRNA interference according to Lipofectamine RNAiMAX instructions (Thermo Fisher Scientific, Waltham, MA, USA) at a final concentration of 100 nM. Briefly, siRNA-ADCY5 or siRNA-NC was transfected into GCs for 48 h. Then, the cells were collected for RNA and protein extraction or for proliferation assays. The sequences of siRNAs used were as follows (5′ to 3′): sense GGACAAGAAUGCCCAGGAATT and antisense UUCCUGGGCAUUCUUGUCCTT; the sequences of negative control (NC) siRNAs used were synthesized as follows: sense UUCUCCGAACGUGUCACGUTT, antisense ACG-UGACACGUUCGGAGAATT.

In order to obtain the CD sequence of *ADCY5*, total RNA was extracted from goat tissue and reverse-transcribed into cDNA using the total template RNA. The amplification of the *ADCY5* CDs region was conducted using specific primers (Table 3), and the PCR products were identified through agarose gel. The PCR products were recovered and inserted into the T vector. Subsequently, the ligated T vector with the inserted products was transformed into *E. coli* DH5α competent cells, cultured at 37 °C for 12 h. Single colonies were picked and cultured in a 50 mL centrifuge tube overnight in a shaker at 160 RPM at 37 °C, followed by plasmid extraction and sequencing by Sangon Biotech Company (Shanghai, China). The amplified DNA fragments were inserted into the pc-DNA3.1 (+) vector to produce the pc-DNA3.1 (+)-ADCY5 plasmid. Subsequently, the constructs were sequenced to confirm their identity. The empty vector served as the control in the experiment. GCs were seeded into 6-well plates or 96-well plates for overexpression of *ADCY5*. Subsequently, upon reaching 70% confluence, transfection was conducted according to the Lipofectamine 3000 manual (Thermo Fisher Scientific) for overexpression plasmid pc-DNA3.1 (+)-ADCY5 or as a control using pc-DNA3.1 (+). After 48 h of transfection, cells were collected for RNA and protein extraction or for proliferation assays.

### 4.8. 8-Br-cAMP Adding Assay

GCs were seeded with 6-well plates or 96-well plates and upon reaching 70% confluence, 100 μM of 8-Bromoadenosine 3′, 5′-cyclic monophosphate (8-Br-cAMP) (AbMole, Houston, TX, USA) was added to the cell culture medium, and cells were collected or cell proliferation detected after 24 h.

### 4.9. Real-Time Quantitative PCR

Total RNA was extracted using Trizol reagent (Invitrogen), and then the purity and concentration of the extracted RNA were assessed. 1000 ng of RNA was transferred to cDNA using PrimeScript™ RT reagent Kit with gDNA Eraser (Takara). The sequences and Gen-Bank accession numbers of primers used to amplify the target genes are shown in Table 3. SYBR Green PCR Master Mix (Takara) was used for quantitative detection of the target genes. Analysis of the relative expression of genes was performed using the 2^−ΔΔCt^ method. β-actin was selected as a reference gene.

### 4.10. Western Blot Analysis

The PIRA cell lysis buffer containing 1× protease inhibitor and 1× phosphatase inhibitor was used to lyse GCs. We determined protein concentration in protein samples using the BCA method (Beyotime, Shanghai, China). To denature the protein samples, we mixed the protein sample loading buffer (Epizyme, Shanghai, China) with a 1/4 volume of the protein sample and boiled them in a water bath for 10 min.

The protein samples were separated by SDS-PAGE on a 12% SDS-polyacrylamide gel and subsequently transferred to a polyvinylidene difluoride (PVDF) membrane. After that, the PVDF membrane was blocked with 5% non-fat milk for 2 h at room temperature, followed by overnight incubation at 4 °C with the primary antibody diluted in antibody dilution buffer (Dongxisw, Xi’an, China). The PVDF membrane was washed three times with tris-buffered saline containing Tween 20 (TBST) for 5 min each. Subsequently, it was incubated with HRP-conjugated goat anti-rabbit/mouse IgG antibody at room temperature. After incubation, the PVDF membrane was washed three times with TBST for 5 min each. Finally, the membranes were stripped using ECL light-emitting liquid (Bio-Rad, Hercules, CA, USA) for exposure. All antibody information is available in Table 4.

### 4.11. Cell Proliferation Assay

The CCK-8 assay was performed using the CCK-8 kit (Beyotime). GCs were seeded into 96-well plates, and each well had 100 μL of medium, and 10 μL of CCK-8 reagent was added to each well. After incubation in a 37 °C constant temperature culture chamber for 1 h, we measured the absorbance at 450 nm to calculate the cell proliferation rate.

The EdU cell proliferation assay (EdU) used the EdU kit (Beyotime). In brief, GCs were seeded in culture dishes, and EdU was added to the dishes (dilution ratio 1:1000) following the instructions and incubated for 3 h. The GCs were then fixed with 4% paraformaldehyde and stained with fluorescent dye and Hoechst [46].

### 4.12. Elisa Assay

We collected cell culture medium and used the ELISA kit (Fankew, Shanghai, China) according to the manufacturer’s instructions to detect the concentrations of E2 and P4 in the medium. In simple terms, we transferred the cell culture supernatant into a 5 mL centrifuge tube and centrifuged for 10 min to obtain the clarified supernatant. Subsequently, we introduced the standards and the clarified supernatant into a microplate, followed by the addition of the enzyme-labeled antibody working solution. Incubate the mixture at room temperature for a duration of 3 h. After incubation, we performed three washing steps, then added the chromogenic substrate and incubated again at room temperature for 30 min in the dark. Finally, we introduced the stop solution, mixed it thoroughly, and determined the optical density (OD) value.

### 4.13. RNA-Fluorescence In Situ Hybridization

RNA-FISH was performed using the Biotin Fluorescence in Situ Hybridization Kit for RNA (Beyotime, China). In brief, GCs were seeded on sterile coverslips treated with TC, fixed with 4% paraformaldehyde at 4 °C for 15 min, incubated with the ADCY5 probe (sequence: T/i2FA//i2FA/TACGTAAACAGRGATTCTCCGCA/i2FG//i2FC/, length: 28) at 37 °C for 4 h, then stained with DAPI for 3 min, observed, and photographed under a fluorescence microscope (Olympus-IX73; Olympus, Tokyo, Japan) [47].

### 4.14. Statistical Analyses

The statistical analysis was conducted using SPSS 17.0 software (SPSS Inc., Chicago, IL, USA). Independent sample *t*-tests were employed to compare the two groups. All values are presented as mean ± standard error of mean (SEM). The levels of significance are denoted as follows: * indicates *p* < 0.05, ** indicates *p* < 0.01, and *** indicates *p* < 0.001.

## 5. Conclusions

This study presents evidence that, despite inducing estrus by simulating the light conditions of the breeding season during the non-breeding season, superior follicular development is observed during the breeding season. The *ADCY5* gene emerges as a potential target influencing seasonal follicular development through its regulation of GC biological functions (Figure 8). This research enhances the understanding of seasonal reproduction in dairy goats and supports advancements in the dairy goat industry.

## Figures and Tables

**Figure 1 ijms-26-01622-f001:**
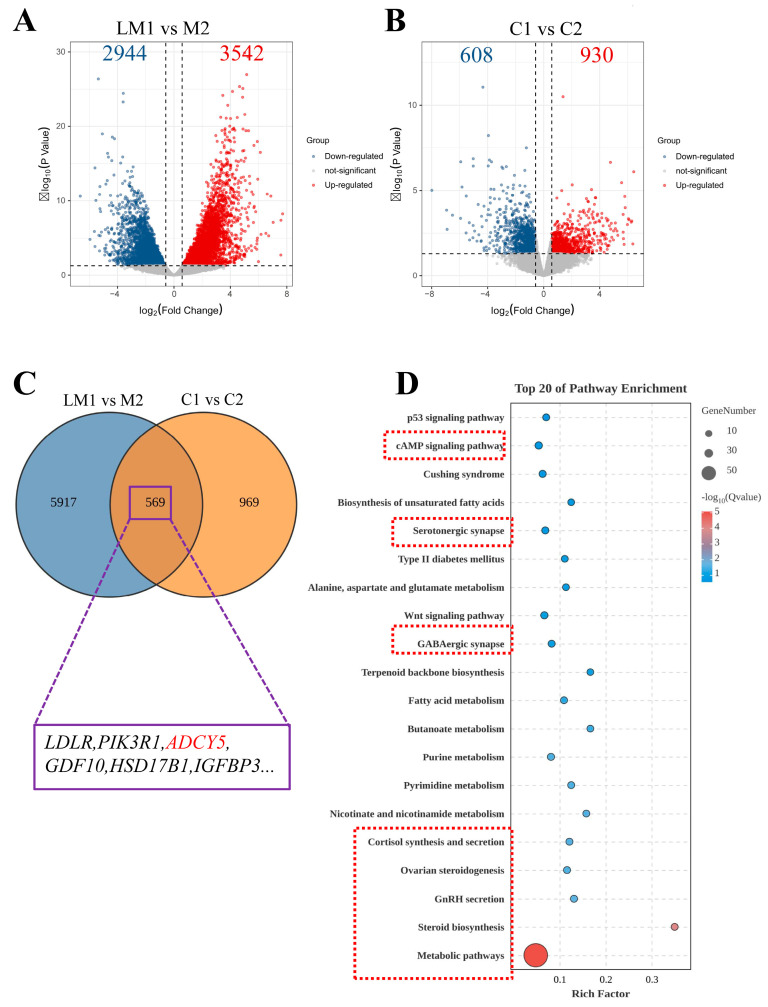
Transcriptome analysis of ovarian tissues. Volcano plot showing DEGs between (**A**) LM1 vs. M2 and (**B**) C1 vs. C2. (**C**) Venn diagram showing distribution of differential genes in LM1 vs. M2 and C1 vs. C2 comparisons. A total of 569 shared DEGs are identified in these comparisons. (**D**) The pathways in KEGG enrichment analysis of the shared DEGs. The red boxes show the pathways related to ovarian hormone synthesis, metabolism, and neurology.

**Figure 2 ijms-26-01622-f002:**
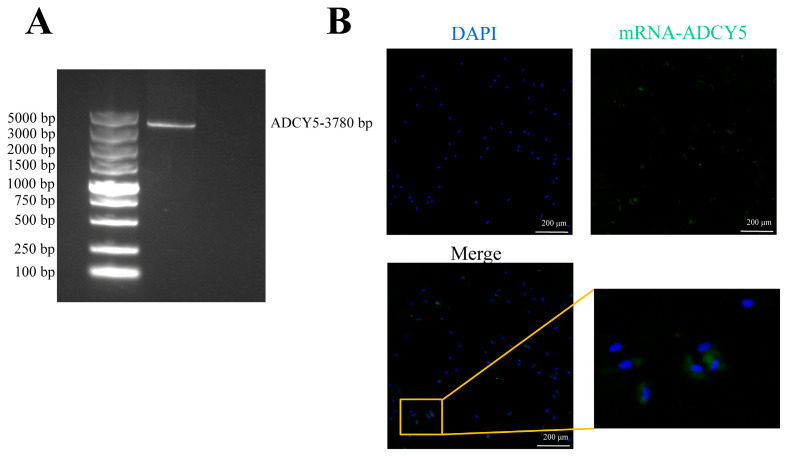
Cloning of the *ADCY5* CDS region and its expression in GCs. (**A**) cDNA cloning of *ADCY5*. (**B**) RNA FISH detection of *ADCY5* expression in GCs (scale bars, 200 µm). DAPI: blue, mRNA-*ADCY5*: green.

**Figure 3 ijms-26-01622-f003:**
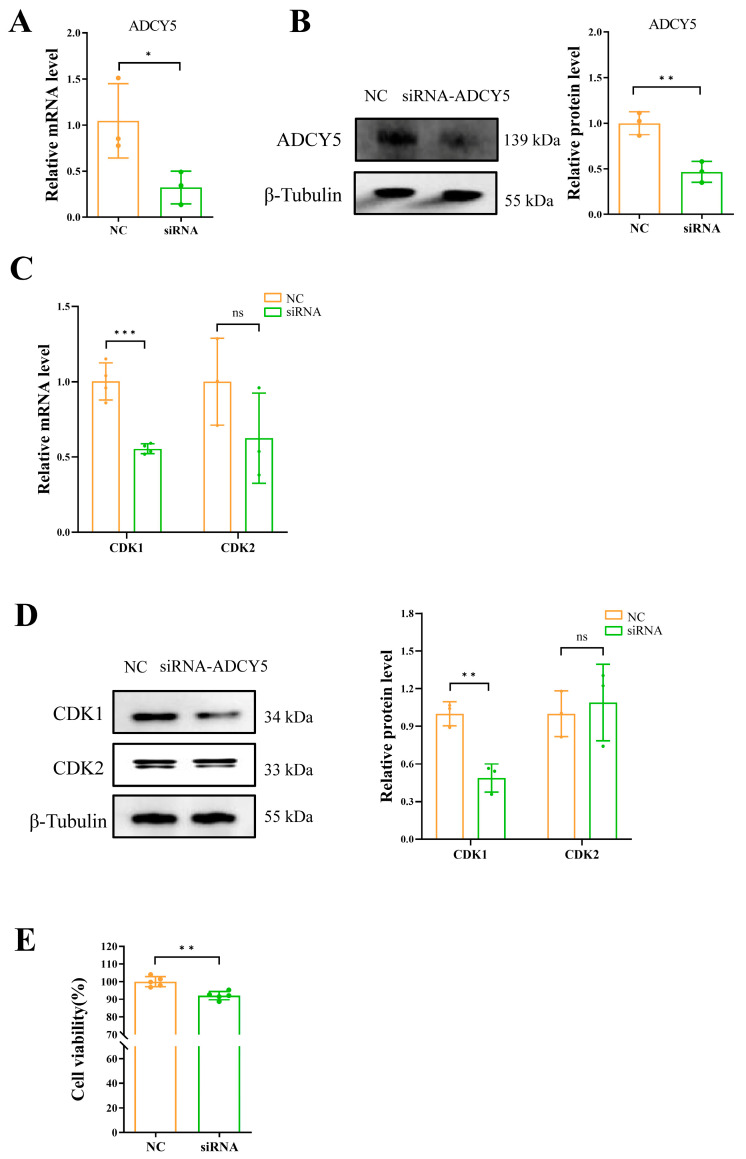
siRNA-mediated knockdown of *ADCY5* reduces the proliferation activity of GCs. The GCs were transfected with or without siRNA. After 48 h of transfection, the proliferation activity was performed to detected. (**A**) The relative mRNA expression levels of *ADCY5* after siRNA interference. (**B**) The protein expression levels of ADCY5 after siRNA interference. (**C**) The relative mRNA level of *CDK1* and *CDK2*. (**D**) The protein expression levels of CDK1 and CDK2. (**E**) Cell viability analysis by CCK8 assay. ns represents *p* ≥ 0.05, * represents *p* < 0.05, ** represents *p* < 0.01, *** represents *p* < 0.001.

**Figure 4 ijms-26-01622-f004:**
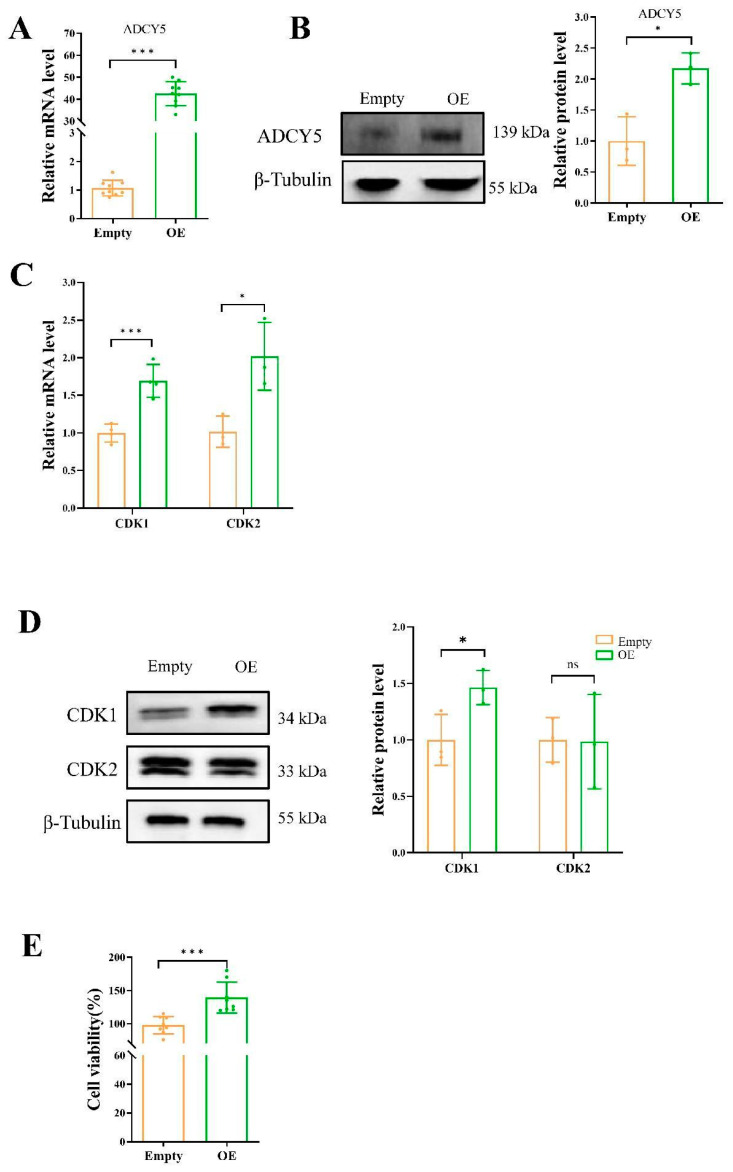
Overexpression of *ADCY5* increases the proliferation activity of GCs. The GCs were transfected with pcDNA-OE or pcDNA-empty. After 48 h of transfection, the proliferation activity was performed to detected. (**A**) The relative mRNA expression levels of *ADCY5* after pcDNA interference. (**B**) The protein expression levels of ADCY5 after pcDNA interference. (**C**) The relative mRNA level of *CDK1* and *CDK2*. (**D**) The protein expression levels of CDK1 and CDK2. (**E**) Cell viability analysis by CCK8 assay. ns represents *p* ≥ 0.05, * represents *p* < 0.05, *** represents *p* < 0.001.

**Figure 5 ijms-26-01622-f005:**
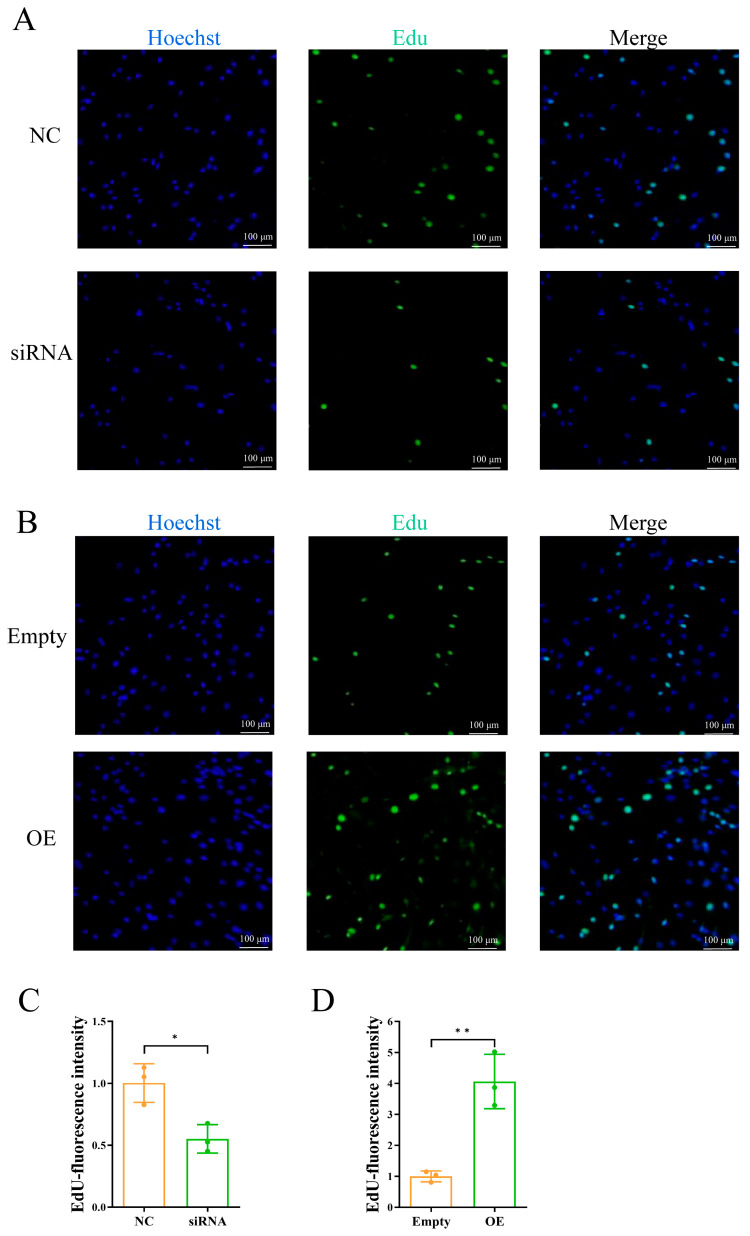
EdU cell proliferation assay. (**A**) Cell proliferation after siRNA interference. (**B**) Cell proliferation after overexpression. (**C**) EdU-fluorescence intensity after siRNA interference. (**D**) EdU-fluorescence intensity after overexpression. Blue: Hoechst. Green: Edu. Scale bar, 100 μm. * represents *p* < 0.05, ** represents *p* < 0.01.

**Figure 6 ijms-26-01622-f006:**
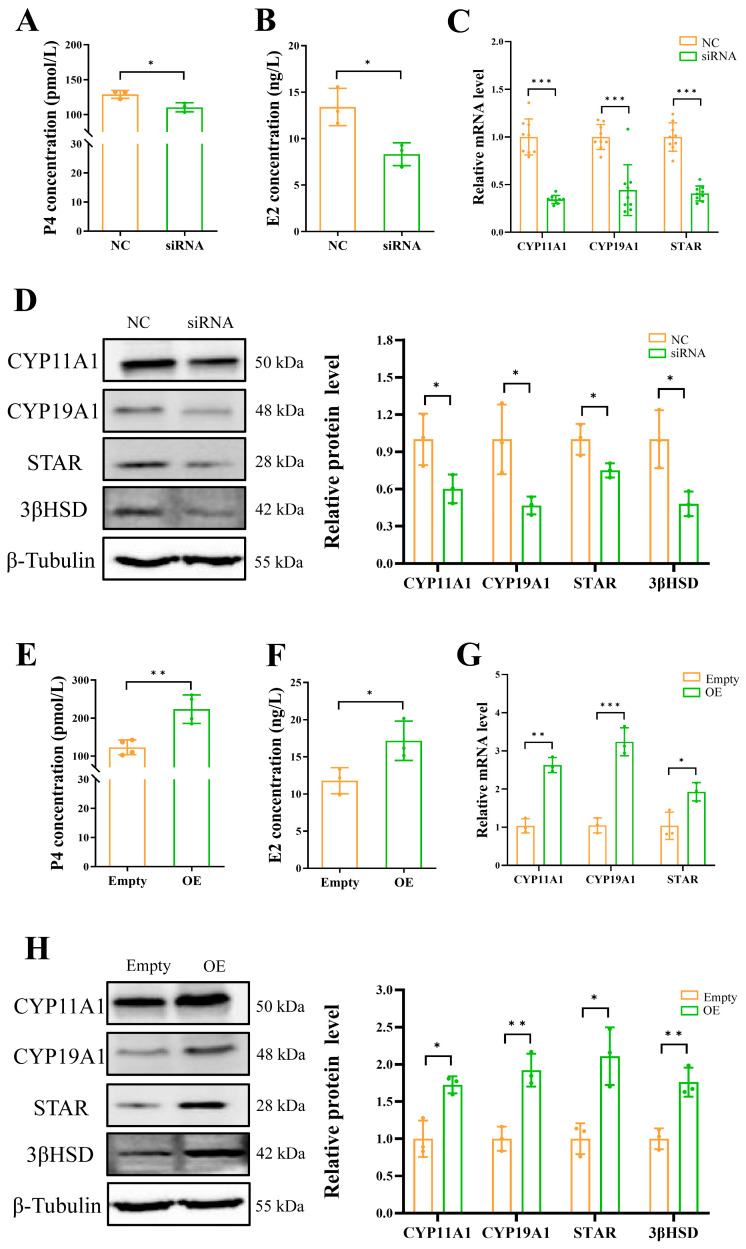
*ADCY5* is involved in regulating the steroidogenesis of GCs. (**A**) Levels of P4 in the culture medium of GCs after *ADCY5* interference. (**B**) Levels of E2 in the culture medium of GCs after *ADCY5* interference. (**C**) mRNA expression levels of *CYP11A1*, *CYP19A1*, and *StAR* in GCs after *ADCY5* interference. (**D**) Protein expression levels of CYP11A1, CYP19A1, StAR, and 3βHSD after *ADCY5* interference. (**E**) Levels of P4 in the culture medium of GCs after *ADCY5* interference. (**F**) Levels of E2 in the culture medium of GCs after *ADCY5* overexpression. (**G**) mRNA expression levels of *CYP11A1*, *CYP19A1*, and *StAR* in GCs after *ADCY5* overexpression. (**H**) Protein expression levels of CYP11A1, CYP19A1, StAR, and 3βHSD after ADCY5 overexpression. * represents *p* < 0.05, ** represents *p* < 0.01, *** represents *p* < 0.001.

**Figure 7 ijms-26-01622-f007:**
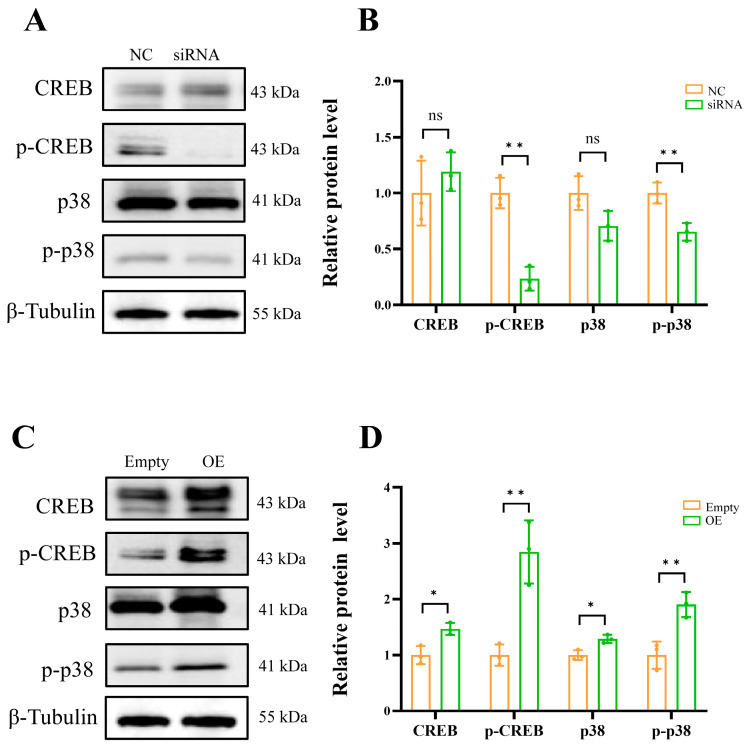
ADCY5-activated CREB activity. (**A**) Protein expression levels and phosphorylation levels of CREB and p38 in GCs after *ADCY5* interference. (**B**) The grayscale analysis of CREB, p38, p-CREB, and p-p38 protein levels after *ADCY5* interference. (**C**) Protein expression levels and phosphorylation levels of CREB and p38 in GCs after *ADCY5* overexpression. (**D**) The grayscale analysis of CREB, p38, p-CREB, and p-p38 protein levels after *ADCY5* overexpression. ns represents *p* ≥ 0.05, * represents *p* < 0.05, ** represents *p* < 0.01.

**Figure 8 ijms-26-01622-f008:**
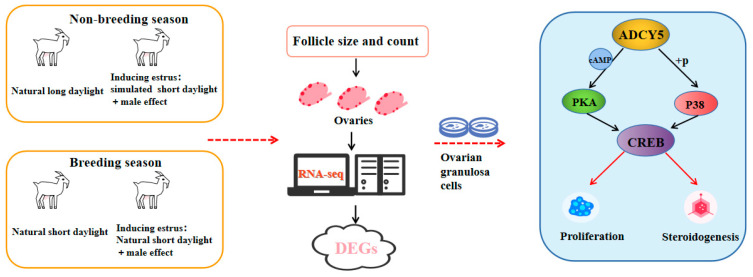
Schematic of proposed model. We observed differences in follicle development after inducing estrus in dairy goats during the breeding and non-breeding seasons. Ovaries were collected for RNA sequencing, and differentially expressed genes (DEGs) were identified. In vitro experiments indicated that the *ADCY5* gene regulates the proliferation of ovarian granulosa cells (GCs) and steroidogenesis through the PKA/CREB and p38 MAPK/CREB pathways.

**Table 1 ijms-26-01622-t001:** The size and count of ovarian follicles.

Items	Experiment 1		Experiment 2		*p*-Value
	Non-Breeding Season		Breeding Season		
Small-sized	C1	7.25 ± 1.71	C2	2.67 ± 1.15	0.011
	LM1	7.00 ± 1.41	M2	3.67 ± 1.53	0.031
Large-sized	C1	0.75 ± 0.96	C2	4.00 ± 2.00	0.034
	LM1	3.25 ± 1.50	M2	11.67 ± 1.15	0.00048
Total	C1	8.00 ± 1.15	C2	6.67 ± 1.15	0.191
	LM1	10.25 ± 1.89	M2	15.33 ± 2.08	0.02

**Table 2 ijms-26-01622-t002:** The seasons, grouping, the body weight, and average feed intake.

Experiments	Seasons	Grouping	Body Weight (kg)	Number	Average Feed Intake (Air-Dry Basis, kg)
Experiment 1	Non-breeding season (March to May)	C1: natural long daylight (control)	46.18 ± 6.97	20	1.18 ± 0.08
		LM1: simulated short daylight and male effect	47.03 ± 6.54	20	1.20 ± 0.06
Experiment 2	Breeding season (August to October)	C2: natural short daylight (control)	48.98 ± 8.09	20	1.16 ± 0.02
		M2: natural short daylight and male effect	47.75 ± 8.31	20	1.28 ± 0.05

**Table 3 ijms-26-01622-t003:** The information of primers.

Gene Name	Primer Sequence (5′-3′)	Product Size (bp)	Accession Number
*CDK1*	F: CCAATAATGAAGTGTGGCCAGAAG	164	XM_054367252.1
	R: AGAAATTCGTTTGGCAGGATCATAG		
*CDK2*	F: AACAAGTTGACGGGAGAAG	237	NM_052827.4
	R: AAGAGGAATGCCAGTGAGT		
*StAR*	F: GGGGATGAGGTGCTGAGTAA	163	XM_013975437.2
	R: TCTGCAGGACCTTGATCTCC		
*CYP11A1*	F: TGGAGGATGTCAAGGCCAAT	239	NM_001287574.1
	R: CACGGAGATAGGGTGGAGTC		
*CYP19A1*	F: ACCAGGTCCCAGCTACTTTC	246	XM_013967046.2
	R: TCATGCATGCCGATGAACTG		
*ADCY5*	F: AGTTCCCATCGGACAAGCTG	198	XM_018045751.1
	R: CGGCCATAACGAGGATCACA		
*ADCY5-CDs*	F: ATGTCCAGCTCCAAAAGCGTGAG	3780	XM_018045751.1
	R: CTAACTGGGCGGGGGCCCTCCGTTGAGGAA		
*β-actin*	F: GGACTTCGAGCAGGAGATGG	140	NM_001314342.1
	R: CCAGGAAGGAAGGCTGGAAG		

**Table 4 ijms-26-01622-t004:** The antibody information.

Antibodies	Cat No.	Source	Dilution
CDK1	310007	Zen-bio, Chengdu, China	1:1000
CDK2	10122-1-AP	Proteintech, Wuhan, China	1:1000
ADCY5	Ab66037	Abcam, Cambridge, UK	1:1000
StAR	A16432	Abclonal, Woburn, MA, USA	1:1000
CYP11A1	A16363	Abclonal	1:1000
CYP19A1	A12684	Abclonal	1:1000
3βHSD	A8035	Abclonal	1:1000
β-Tubulin	YM3030	Immunoway, Plano, TX, USA	1:1000
CREB	9197	Cell Signaling, Danvers, MA, USA	1:1000
p-CREB	9198	Cell Signaling	1:1000
p38	14064-1-AP	Proteintech	1:1000
p-p38	AF4001	Affinity Biosciences, Cincinnati, OH, USA	1:1000
Goat anti-Rabbit IgG HRP Conjugated	CW0103S	Cwbio, Taizhou, China	1:5000
Goat anti-Mouse IgG HRP Conjugated	CW0102S	Cwbio	1:5000

## Data Availability

The original contributions presented in the study are included in the article/Appendix A, further inquiries can be directed to the corresponding authors.

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
