# Peer review of "ADCY5 Gene Affects Seasonal Reproduction in Dairy Goats by Regulating Ovarian Granulosa Cells Steroid Hormone Synthesis"

_ijms, 2025, doi:10.3390/ijms26041622_

Round 1
Reviewer 1 Report
Comments and Suggestions for Authors
The author mined the differentially expressed genes by analyzing the transcriptomes of ovaries at different seasons. Then, the gene function of ADCY5 was studied in vitro, and the mechanism was further explored. This study contributes to a better understanding of the seasonal reproductive variations in dairy goats and the potential role of the ADCY5 gene in the seasonal reproductive processes of dairy goats.
Overall, the data are sufficient, but major revision is needed before publication.
Line 15, 'follicular size differences', This study is more like the differences of ovaries at different development stages, in which the proportion of different size follicles is different. Is it accurate here?
Line 18-20, 'larger follicles'; 'size differences'. Revise this sentence.
'ovarian tissues' should be revised to 'ovaries'.
Line 20-21, Our analysis revealed…….
Line 38-40, The induction procedures using exogenous…….
The language is inaccurate or redundant. The whole text should be revised for better readability.
Line 50-51, P4, PMSG, and PGF2α. Provide the full name when they first appear.
Lines 69-70 and 84, What is the meaning of C1, C2, LM1, M1, need to introduce the experimental grouping briefly.
Line 87-94, 'The RNA-Seq analysis uncovered…… (Figure 1B and Table S1), …… (Figure 1A and Table S1).' The description of Fig.1A should appear before Fig.1B.
Add the threshold for these DEGs here.
List the names or IDs of DEGs, which might be helpful for readers.
Lines 115-116, Fig2A, lacks a brief DNA ladder definition.
Fig2B, To improve the figure picture resolution. Moreover, I think it is hard to distinguish cell types here. Can you provide the figure of the two-color in situ hybridization of ADCY5 and a well-known GC maker?
And control, the result of sense-probe, is needed.
Line 110, 2.3 ADCY5 is involved in regulating the proliferation of GCs.
This section is haphazard. I suggest you re-write this section or rearrange the Figures.
Line 136 and 151, Can you provide a statistical of fluorescence intensity in Fig 4?
Line 156, E2 should be revised to Estradiol (E2). Provide the full name of Abbreviations if it is the first time it appears. Please check.
Line 158, Figures 5C and D should be Figures 5F and G. Please check.
I suggest describing all the results after knockdown ADCY5 first and then describing the results of overexpression.
Lines 200 and 202, add the Latin names of single-humped camels and Boer goats if it is the first time it appears.
Line 307, please check the punctuation.
Line 386, Table 2 should be revised to Table 3.
Line 387, add 'β-actin was selected as a reference gene'
Line 418, Briefly introduce the protocol of Elisa assay.
Line 428-429, Which section in the manuscript needs to use one-way ANOVA?
Comments on the Quality of English LanguageThe language is inaccurate or redundant. The whole text should be revised for better readability.
Author Response
Comments 1: [Line 15, 'follicular size differences', This study is more like the differences of ovaries at different development stages, in which the proportion of different size follicles is different. Is it accurate here?]
|
Response 1:[This study investigated follicle development during the breeding and non-breeding seasons and explored molecular mechanisms for variations in the proportions of follicles of different sizes using ovarian RNA-seq and in vitro experiments.] Thank you for pointing this out. We agree with this comment. Therefore, we have counted the number of antral follicles, and since “differences in follicle size” seemed insufficiently accurate, we modified it to “differences in the proportion of follicles of different sizes”.Changes have been made in the manuscript, see lines 13-16. |
Comments 2: [Line 18-20, 'larger follicles'; 'size differences'. Revise this sentence. 'ovarian tissues' should be revised to 'ovaries'.] |
Response 2: [This study identified an increase in follicle size during the breeding season and performed RNA-seq on ovaries to explore the underlying causes.]Thank you for pointing this out. We agree with this comment. Therefore, we have made modifications.Changes have been made in the manuscript, see lines 18-20. |
Comments 3: [Line 20-21, Our analysis revealed…….] |
Response 3: [The RNA-seq analysis elucidated pathways associated with cellular and hormonal metabolism, and identified adenylyl cyclase 5 (ADCY5) as a key differentially expressed gene.]Thank you for pointing this out. We agree with this comment. Therefore, we have made modifications.Changes have been made in the manuscript, see lines 20-22. |
Comments 4: [Line 38-40, The induction procedures using exogenous……. The language is inaccurate or redundant. The whole text should be revised for better readability.] |
Response 4: [Various estrus induction protocols are widely used in production, including the use of progesterone sponges in conjunction with prostaglandin F2α (PGF2α), which have proven effective in inducing estrus[1]. Additionally, the combination of melatonin (MEL) and progesterone (P4) has been shown to successfully induce estrus and ovulation, thereby enhancing reproductive success rates [2]. Beyond exogenous hormone treatments, behav-ioral stimuli are also employed to induce estrus. For instance, the introduction of sexually active male goats to anestrous females during the non-breeding season can effectively stimulate estrus and ovulation through male-female interactions, a phenomenon referred to as the "male effect[3]." As short-day breeders, dairy goats reproductive activities are in-fluenced by photoperiod. Introducing a short photoperiod is another method for inducing estrus; for example, exposing ewes to artificial long-day lighting followed by a natural photoperiod can effectively induce estrus in ewes during the non-breeding season[4]. ]Thank you for pointing this out. We agree with this comment. Therefore, we have rewritten this paragraph to improve readability. Changes have been made in the manuscript, see lines 42-54. |
Comments 5: [Line 50-51, P4, PMSG, and PGF2α. Provide the full name when they first appear.] |
Response 5: [progesterone,Pregnant mare serum gonadotropin and prostaglandin F2α]Thank you for pointing this out. We agree with this comment. Therefore, we have added the full names for the hormones when they first appear.Changes have been made in the manuscript, see lines 44-63. |
Comments 6: [Lines 69-70 and 84, What is the meaning of C1, C2, LM1, M1, need to introduce the experimental grouping briefly.] |
Response 6: [In Experiment 1(non-breeding season), there were 20 dairy goats under natural long-day light treatment (control group,C1 group) and 20 dairy goats that received simulated short-day light mimicking the breeding season along with male effect treatment(LM1 group).In Experiment 2(breeding season), there were 20 dairy goats under natural short-day light treatment (control group, C2 group) and 20 dairy goats receiving both natural short-day light and male effect treatment(M2 group). ] Thank you for pointing this out. We agree with this comment. Therefore, we have briefly introduced the experimental grouping information at the beginning of the first result.Changes have been made in the manuscript, see lines 83-89. |
Comments 7: [Line 87-94, 'The RNA-Seq analysis uncovered…… (Figure 1B and Table S1), …… (Figure 1A and Table S1).' The description of Fig.1A should appear before Fig.1B. Add the threshold for these DEGs here. List the names or IDs of DEGs, which might be helpful for readers.] |
Response 7: [To investigate the molecular mechanisms influencing follicle development, we performed transcriptome sequencing of the ovaries. The RNA-Seq analysis uncovered 3542 up-regulated and 2944 down-regulated DEGs in LM1 vs M2. These DEGs include steroidogenic acute regulatory protein(STAR, |log2FC|=1.93), 3 beta-hydroxysteroid dehydrogenase(3BHSD, |log2FC|=3.65), cytochrome P450 family 11 subfamily A member 1(CYP11A1, |log2FC|=2.54), epidermal growth factor receptor(EGFR, |log2FC|=1.21), ADCY5(|log2FC|=2.06), AKT serine/threonine kinase 3(AKT3, |log2FC|=3.21), and mitogen-activated protein kinase 12(MAPK12, |log2FC|=1.14)(Fig.1A and Supplement Table 1). In C1 vs C2, there were 930 up-regulated and 608 down-regulated DEGs. These DEGs include STAR(|log2FC|=1.01), ADCY5(|log2FC|=0.59), bone morphotic protein 15(BMP15, |log2FC|=0.89), follicle stimulating hormone receptor(FSHR, |log2FC|=3.37), and Cytochrome P450 17A1(CYP17A1,|log2FC|=3.17) (Fig.1B and Supplement Table 1). There are 569 shared DEGs, including low-density lipoprotein receptor(LDLR),phosphoinositide-3-kinase regulatory subunit 1(PIK3R1), ADCY5, growth differentiation factor 10(GDF10), hydroxysteroid dehydrogenase 17 beta 1(HSD17B1),insulin-like growth factor binding protein 3(IGFBP3)(Fig.1C). ] Thank you for pointing this out. We agree with this comment. Therefore, we have changed the order of the figures and added the full names and thresholds for the DEGs.Changes have been made in the manuscript, see lines 105-120. |
Comments 8: [Lines 115-116, Fig2A, lacks a brief DNA ladder definition.] |
Response 8: [ Figure 2A.] Thank you for pointing this out. We agree with this comment. Therefore, we have added the DNA marker labels in Figure 2A. Changes have been made in the manuscript, see Figure 2. |
Comments 9: [Fig2B, To improve the figure picture resolution. Moreover, I think it is hard to distinguish cell types here. Can you provide the figure of the two-color in situ hybridization of ADCY5 and a well-known GC maker? And control, the result of sense-probe, is needed.] |
Response 9: [we have uploaded high-resolution tif format images, explained the reason for not adding markers, and presented images of the positive control]Thank you for pointing this out. We agree with this comment. Therefore, we have uploaded high-resolution TIF format images. GCs, as a common cell type, typically use FSHR as a specific marker in protein immunofluorescence. GCs are present in the collected follicular fluid, and there may also be oocytes. Considering the large size of the oocytes, we filtered the cells using a 200-mesh filter, allowing us to conclude that the collected GCs were not mixed with other cell types. In the RNA-FISH experiment, we included GAPDH as a positive control, but since it is uncommon to display the positive control in the results, we did not include images of the positive control. Below are the images of the positive control.
|
Comments 10: [Line 110, 2.3 ADCY5 is involved in regulating the proliferation of GCs. This section is haphazard. I suggest you re-write this section or rearrange the Figures.] |
Response 10: [Results 2.3 was rewritten]Thank you for pointing this out. We agree with this comment. Therefore, we have rewritten this section and rearranged the figures.Changes have been made in the manuscript, see lines 168-219. |
Comments 11: [Line 136 and 151, Can you provide a statistical of fluorescence intensity in Fig 4?] |
Response 11: [Figure 5C andD]Thank you for pointing this out. We agree with this comment. Therefore, we have added a statistical result of the fluorescence intensity.Changes have been made in the manuscript, see Figure 5C andD. |
Comments 12: [Line 156, E2 should be revised to Estradiol (E2). Provide the full name of Abbreviations if it is the first time it appears. Please check.] |
Response 12: [estradiol(E2)] Thank you for pointing this out. We agree with this comment. Therefore, we have made revisions.Changes have been made in the manuscript, see lines 322-323. |
Comments 13: [Line 158, Figures 5C and D should be Figures 5F and G. Please check. I suggest describing all the results after knockdown ADCY5 first and then describing the results of overexpression.] |
Response 13: [Changes have been made in the manuscript ]Thank you for pointing this out. We agree with this comment. Therefore, we have made revisions.Changes have been made in the manuscript, seeFigure 6. |
Comments 14: [Lines 200 and 202, add the Latin names of single-humped camels and Boer goats if it is the first time it appears.] |
Response 14: [ (Capra hircus ) and (Camelus dromedarius)]Thank you for pointing this out. We agree with this comment. Therefore, we have added the Latin names and changed “single-humped camels” to “dromedary camels”.Changes have been made in the manuscript, see lines 392-395. |
Comments 15: [Line 307, please check the punctuation.] |
Response 15: [The punctuation has been changed to “.”] Thank you for pointing this out. We agree with this comment. Therefore, we have modified the punctuation.Changes have been made in the manuscript, see lines 523. |
Comments 16: [Line 386, Table 2 should be revised to Table 3.] |
Response 16: [The sequences and Gen-Bank accession numbers of primers used to amplify the target genes are shown in Table 3.]Thank you for pointing this out. We agree with this comment. Therefore, we have changed the Table2 to Table 3.Changes have been made in the manuscript, see lines597-599. |
Comments 17: [Line 387, add 'β-actin was selected as a reference gene'] |
Response 17: [β-actin was selected as a reference gene]Thank you for pointing this out. We agree with this comment. Therefore, we have added this sentence.Changes have been made in the manuscript, see lines 601. |
Comments 18: [Line 418, Briefly introduce the protocol of Elisa assay.] |
Response 18: [In simple terms, transfer the cell culture supernatant into a 5 mL centrifuge tube and centrifuge for 10 min to obtain the clarified supernatant. Subsequently, introduce the standards and the clarified supernatant into a microplate, followed by the addition of the enzyme-labeled antibody working solution. Incubate the mixture at room temperature for a duration of 3 h. After incubation, perform three washing steps, then add the chromogenic substrate and incubate again at room temperature for 30 min in the dark. Finally, introduce the stop solution,] Thank you for pointing this out. We agree with this comment. Therefore, we have provided details regarding the ELISA.Changes have been made in the manuscript, see lines 629-637. |
Comments 19: [Line 428-429, Which section in the manuscript needs to use one-way ANOVA?] |
Response 19: [The statistical analysis was conducted using SPSS 17.0 software (SPSS Inc., Chicago, IL, USA). Independent sample t-tests were employed to compare the two groups. All values are presented as mean ± standard error of mean (S.E.M.). The levels of significance are denoted as follows: * indicates P < 0.05, ** indicates P < 0.01, and *** indicates P < 0.001.]Thank you for pointing this out. We agree with this comment. Therefore, we have rewritten this section, as we did not use one-way ANOVA but instead only employed independent samples t-test.Changes have been made in the manuscript, see lines 646-649. |

Reviewer 2 Report
Comments and Suggestions for Authors
ijms-3421305-peer-review-report-v1. Thank you for the opportunity to review this paper. It is well-written, citing current research in this area. The manuscript, even though informative, could benefit from several improvements in clarity, precision, and flow. Here are some specific areas that require revision: Abstract a) The oestrus induction techniques, like “simulated breeding season short photoperiod” and “male effect method,” are unclear. Brief descriptions of these terms would clarify their application, especially for non-specialist readers who may be unused to these. b) The detection of ADCY5 as a significant gene seems hasty. To clarify its significance and context, the selection process and its importance to follicle development and reproductive physiology in goats should be explained. Introduction a) Lines 38-39: The phrase “exogenous exogenous hormones” is repeated. It could be simplified to just “exogenous hormones.” b) Lines 62-64: This sentence should be paraphrased for clarity. Suggestion: We hypothesise differences in follicular development in dairy goats between the breeding and non-breeding seasons, and this study aims to investigate these differences. c) The introduction alternates between references to "breeding" and "nonbreeding" seasons without a clear distinction between what these seasons imply for goats, ewes, or sheep. A short description of these terms at the start would increase reader understanding. d) Abbreviations like P4, P4+PMSG, and PGF2α are used without being spelt out or briefly explained. Authors should give the full names with abbreviations in parentheses upon first use, e.g., "progesterone (P4)," "pregnant mare serum gonadotropin (PMSG)." e) The introduction ends hastily with the study aims. A brief summary stating the knowledge gaps that the study seeks to address would help link the background to the objectives more smoothly. Discussion a) Line 203: Remove the phrase “reason for the”. It now becomes “The slow development of these non-reproductive stage…. b) Line 211: Change the wordings “…breeding season and non-breeding season.” It becomes “…breeding and non-breeding seasons.” c) Line 217: Add a hyphen between “Progesterone” and “mediated” (Progesterone-mediated). d) The transition from observing follicle size to discussing molecular mechanisms feels hasty. Introducing a brief linking sentence to clarify how the observed differences in follicle size relate to the molecular findings would help. e) The discussion section ends by suggesting that ADCY5 regulates steroidogenesis, but a stronger concluding statement that ties together the implications of these findings would be ideal. It would be better to summarize the broader significance of the research, especially in terms of improving reproductive management in dairy goats. Materials and Methods a) The formatting for experiment names is inconsistent. At the start of the section, "Experiment 1" and "Experiment 2" are not defined clearly in terms of objectives. The authors should furnish more context or a clearer distinction in the early part of the section. b) The rationale behind the gradual reduction of light exposure (e.g., 15 minutes weekly) and its specific choice needs a clearer explanation. Also, more details are required on the "male effect" protocol, particularly its impact on oestrus induction in dairy goats versus other methods. c) Though methods like RNA extraction, Western blot, and RT-PCR are wellexplained, they could be streamlined. These are standard techniques in molecular biology, so the section could reference established protocols or earlier studies to reduce excessive detail and keep the focus on key aspects of the research. d) The addition of 100 μM 8-Br-cAMP is mentioned, but its exact role in the experiment is not fully explained. A brief explanation of how 8-Br-cAMP acts within the context of ADCY5 overexpression would enhance understanding. Additionally, how its effects on cell proliferation or other molecular pathways were quantified would improve clarity. e) While the study describes light treatments and the male effect in detail, it could benefit from discussing potential confounding variables. For example, other environmental factors (e.g., temperature, humidity, feed variations) may affect the goats' reproductive cycles. f) The ADCY5 gene’s role in granulosa cell proliferation is well-covered, but other hormones like FSH, LH, and estradiol, which also influence follicular development, are not addressed. Briefly mentioning how these hormones were measured or controlled would provide valuable context for the hormonal aspects of the experiments. Conclusions a) The mention of "ovarian transcriptome sequencing and in vitro experiments" is valuable, but it is somewhat out of place in the conclusion. A conclusion should primarily focus on the implications of the findings rather than reiterating the methods used, which could be summarized earlier in the discussion. b) While the conclusion mentions the role of ADCY5 in seasonal reproductive processes, it doesn’t explicitly address the study's initial hypothesis or objectives. Restating the hypothesis and explaining how the results confirm or refute it would create a stronger sense of closure. c) Figure 7 appears out of place. I doubt if it is suitable for the conclusion or any other section of this paper. It may be better used for literature review manuscripts.

Author Response
Comments 1: [The oestrus induction techniques, like “simulated breeding season short photoperiod” and “male effect method,” are unclear. Brief descriptions of these terms would clarify their application, especially for non-specialist readers who may be unused to these] |
Response 1: [Beyond exogenous hormone treatments, behavioral stimuli are also employed to induce estrus. For instance, the introduction of sexually active male goats to anestrous females during the non-breeding season can effectively stimulate estrus and ovulation through male-female interactions, a phenomenon referred to as the "male effect[3]." As short-day breeders, dairy goats reproductive activities are influenced by photoperiod. Introducing a short photoperiod is another method for inducing estrus; for example, exposing ewes to artificial long-day lighting followed by a natural photoperiod can effectively induce estrus in ewes during the non-breeding season[4].] Thank you for pointing this out. We agree with this comment. Therefore, we have provided explanations for “short photoperiod” and “male effect” in the introduction to help non-expert readers understand more easily. Changes have been made in the manuscript, see lines 47-54. |
Comments 2: [The detection of ADCY5 as a significant gene seems hasty. To clarify its significance and context, the selection process and its importance to follicle development and reproductive physiology in goats should be explained.] |
Response 2: [The RNA-Seq analysis identified 569 shared DEGs, and upon reviewing the studies of these DEGs, we found that ADCY5 may be associated with the seasonality of animal reproduction[22]. Oocyte maturation is regulated by intracellular cAMP levels through the activity of endogenous adenylyl cyclase, suggesting that ADCY may be involved in the early stages of oocyte meiosis[23]. The whole genome association study (GWAS) identified ADCY5 as a potential gene related to bovine fertility, and its genetic polymorphism affects ovarian width and the diameter of the corpus luteum[24]. ADCY5 may also play a signifi-cant role in regulating seasonal reproduction in sheep[22].] Thank you for pointing this out. We agree with this comment. Therefore, in the discussion, we stated the reason for our selection: ADCY5 is one of the shared DEGs, and there have been studies indicating that it may be related to seasonal reproduction. However, these studies have not further explored the molecular mechanisms, which is why we chose this DEG.Changes have been made in the manuscript, see lines 423-430. |
Comments 3: [Lines 38-39: The phrase “exogenous exogenous hormones” is repeated. It could be simplified to just “exogenous hormones.” ] |
Response 3: [Various estrus induction protocols are widely used in production, including the use of progesterone sponges in conjunction with prostaglandin F2α (PGF2α), which have proven effective in inducing estrus[1].] Thank you for pointing this out. We agree with this comment. Therefore, we have corrected the repetitive wording errors and revised this section to make it more fluent and coherent.Changes have been made in the manuscript, see lines 42-45. |
Comments 4: [ Lines 62-64: This sentence should be paraphrased for clarity. Suggestion: We hypothesise differences in follicular development in dairy goats between the breeding and non-breeding seasons, and this study aims to investigate these differences. ] |
Response 4: [We hypothesise differences in follicular development in dairy goats between the breeding and non-breeding seasons, and this study aims to investigate these differences.] Thank you for pointing this out. We agree with this comment. Therefore, we have made revisions.Changes have been made in the manuscript, see lines 73-75. |
Comments 5: [The introduction alternates between references to "breeding" and "nonbreeding" seasons without a clear distinction between what these seasons imply for goats, ewes, or sheep. A short description of these terms at the start would increase reader understanding. ] |
Response 5: [In the northwestern region of China, located in the Northern Hemisphere, the anestrous period for goats spans from January to June. During this time, the goats experience a prolonged photoperiod and enter a reproductive quiescence, commonly termed the non-breeding season. Conversely, from July to December, the photoperiod shortens, and the goats transition into a reproductive phase known as the breeding season. ] Thank you for pointing this out. We agree with this comment. Therefore, we have made a statement at the beginning of the introduction. Changes have been made in the manuscript, see lines 34-38. |
Comments 6: [Abbreviations like P4, P4+PMSG, and PGF2α are used without being spelt out or briefly explained. Authors should give the full names with abbreviations in parentheses upon first use, e.g., "progesterone (P4)," "pregnant mare serum gonadotropin (PMSG)."] |
Response 6: [progesterone,Pregnant mare serum gonadotropin and prostaglandin F2α]Thank you for pointing this out. We agree with this comment. Therefore, we have added the full names for the hormones when they first appear.Changes have been made in the manuscript, see lines 44-63. |
Comments 7: [The introduction ends hastily with the study aims. A brief summary stating the knowledge gaps that the study seeks to address would help link the background to the objectives more smoothly. ] |
Response 7: [This study aims to provide new insights into the seasonal reproduction differences of dairy goats. The ultimate aim is to enhance the reproductive performance of dairy goats and facilitate continuous milk production throughout the year.] Thank you for pointing this out. We agree with this comment. Therefore, we have made revisions. Changes have been made in the manuscript, see lines 77-80. |
Comments 8: [Line 203: Remove the phrase “reason for the”. It now becomes “The slow development of these non-reproductive stage….] |
Response 8: [The slow development of these non-reproductive stage follicles may be due to the overall lower LH levels during the non-reproductive stage] Thank you for pointing this out. We agree with this comment. Therefore, we have made revisions. Changes have been made in the manuscript, see lines 396-398. |
Comments 9: [Line 211: Change the wordings “…breeding season and non-breeding season.” It becomes “…breeding and non-breeding seasons.”] |
Response 9: [In our study, we conducted ovarian transcriptome analysis to explore the potential molecular mechanisms underlying the differences in follicle development between the breeding and non-breeding season.] Thank you for pointing this out. We agree with this comment. Therefore, we have made revisions. Changes have been made in the manuscript, see lines 408-410. |
Comments 10: [Line 217: Add a hyphen between “Progesterone” and “mediated” (Progesterone-mediated).] |
Response 10:[Progesterone-mediated oocyte maturation ] Thank you for pointing this out. We agree with this comment. Therefore, we have made revisions. Changes have been made in the manuscript, see lines 416. |
Comments 11: [The transition from observing follicle size to discussing molecular mechanisms feels hasty. Introducing a brief linking sentence to clarify how the observed differences in follicle size relate to the molecular findings would help. ] |
Response 11: [The size of ovarian follicles is intrinsically linked to the developmental potential of oocytes, with its regulation being governed by a multitude of molecular mechanisms, such as the regulation of gene expression, signaling pathways and components of follicular fluid. Furthermore, throughout follicular development, the gene expression activities of granulosa cells play a crucial role in sustaining the intricate network of signaling pathways within the follicles.] Thank you for pointing this out. We agree with this comment. Therefore, we have added a transition sentence to clarify the relationship between the observed differences in follicle size and the molecular findings. Changes have been made in the manuscript, see lines 401-407. |
Comments 12: [The discussion section ends by suggesting that ADCY5 regulates steroidogenesis, but a stronger concluding statement that ties together the implications of these findings would be ideal. It would be better to summarize the broader significance of the research, especially in terms of improving reproductive management in dairy goats.] |
Response 12: [Consequently, we propose that ADCY5 modulates CREB activity via the cAMP/PKA and p38 MAPK signaling pathways, thereby affecting steroid hormone synthesis in GCs and facilitating follicular development during the breeding season. This mechanism may elu-cidate the observed increase in pregnancy rates among dairy goats during this period. Our research identifies a potential target for enhancing reproductive performance in dairy goats during the non-breeding season.] Thank you for pointing this out. We agree with this comment. Therefore, we have added a strong conclusion that relates to our findings, which have more significance for the reproductive management of dairy goats. Changes have been made in the manuscript, see lines 469-475. |
Comments 13: [The formatting for experiment names is inconsistent. At the start of the section, "Experiment 1" and "Experiment 2" are not defined clearly in terms of objectives. The authors should furnish more context or a clearer distinction in the early part of the section. ] |
Response 13: [Induction estrus experiments were conducted during the non-breeding season and the breeding season, thus “Experiment 1” and ”Experiment 2” are used to distinguish between the different periods.] Thank you for pointing this out. We agree with this comment. Therefore, we have provided an explanation that “Experiment 1” and “Experiment 2” are intended to distinguish between the different periods. |
Comments 14: [The rationale behind the gradual reduction of light exposure (e.g., 15 minutes weekly) and its specific choice needs a clearer explanation. Also, more details are required on the "male effect" protocol, particularly its impact on oestrus induction in dairy goats versus other methods.] |
Response 14: [Gradually reducing the photoperiod is intended to simulate a short photoperiod. At the experimental site, the natural light exposure during the breeding season decreases by approximately 15 minutes each week, so we chose to reduce the light duration by 15 minutes each week to simulate the short photoperiod.The staff uses a tether to bring the male goats into the female goat pens every morning at 8 a.m. They are allowed to interact and court each other, but cannot mate, and this process lasts for 30 minutes.] Thank you for pointing this out. We agree with this comment. Therefore, we have provided an explanation for ”15 min” and added more details about the “male effect”.Changes have been made in the manuscript, see lines 514-516. |
Comments 15: [Though methods like RNA extraction, Western blot, and RT-PCR are wellexplained, they could be streamlined. These are standard techniques in molecular biology, so the section could reference established protocols or earlier studies to reduce excessive detail and keep the focus on key aspects of the research.] |
Response 15: [Total RNA was extracted using Trizol reagent (Invitrogen, USA), and then the purity and concentration of the extracted RNA were assessed. ] Thank you for pointing this out. We agree with this comment. Therefore, we have simplified the description of qPCR, and the details of RNA extraction and Western blot have also been streamlined. We wanted to retain the main steps to meet the needs of non-expert readers.Changes have been made in the manuscript, see lines 530-531. |
Comments 16: [The addition of 100 μM 8-Br-cAMP is mentioned, but its exact role in the experiment is not fully explained. A brief explanation of how 8-Br-cAMP acts within the context of ADCY5 overexpression would enhance understanding. Additionally, how its effects on cell proliferation or other molecular pathways were quantified would improve clarity. ] |
Response 16: [Furthermore, since ADCY5 can promote the synthesis of cAMP, we employed the PKA activator 8-Br-cAMP, a cell-permeable cAMP analog, to compare its effects with those observed from ADCY5 overexpression.] Thank you for pointing this out. We agree with this comment. Therefore, we have explained this in Results 3.5. 8-Br-cAMP, as a cAMP analog, is widely used in cell proliferation experiments, and we also investigated its effect on GCs proliferation in Supplementary Figure 2. Changes have been made in the manuscript, see lines 344-346. |
Comments 17: [While the study describes light treatments and the male effect in detail, it could benefit from discussing potential confounding variables. For example, other environmental factors (e.g., temperature, humidity, feed variations) may affect the goats' reproductive cycles.] |
Response 17: [Environmental factors may affect the reproductive cycle of goats. Therefore, we mentioned in section 4.1 that the same feed ratio and feeding methods were used, and the average intake amount is noted in Table 2. Temperature and humidity are also factors that can influence the reproductive cycle; however, photoperiod plays a key role in the reproductive cycle of goats. Therefore, photoperiod is usually considered one of the main variables in experiments aimed at inducing reproduction during the non-breeding season, and no separate experiments were designed specifically for temperature and environmental factors.] Thank you for pointing this out. We agree with this comment. Therefore, we have explained this. |
Comments 18: [The ADCY5 gene’s role in granulosa cell proliferation is well-covered, but other hormones like FSH, LH, and estradiol, which also influence follicular development, are not addressed. Briefly mentioning how these hormones were measured or controlled would provide valuable context for the hormonal aspects of the experiments.] |
Response 18: [In this study, the secretion of P4 and E2 by GCs is modulated by ADCY5. P4 priming during follicular development has been demonstrated to enhance oocyte competence, resulting in increased rates of cleavage and embryo development in sheep[34]. Additionally, another investigation examined the impact of estradiol on bovine cumulus-oocyte complexes, revealing that E2 supplementation is essential for preserving oocyte quality and maturity[35].] Thank you for pointing this out. We agree with this comment. In the culture of GCs, we did not add FSH, LH, or E2. We analyzed the ability of GCs to secrete P4 and E2 through ELISA, as both can regulate the maturation of oocytes. Therefore, we addressed this in the discussion. Changes have been made in the manuscript, see lines 444-450. |
Comments 19: [The mention of "ovarian transcriptome sequencing and in vitro experiments" is valuable, but it is somewhat out of place in the conclusion. A conclusion should primarily focus on the implications of the findings rather than reiterating the methods used, which could be summarized earlier in the discussion. ] |
Response 19: [This study presents evidence that, despite inducing estrus by simulat-ing the light conditions of the breeding season during the non-breeding season, superior follicular de-velopment is observed during the breeding season. The ADCY5 gene emerges as a poten-tial target influencing seasonal follicular development through its regulation of GCs biological functions (Fig.8). This research enhances the understanding of seasonal reproduc-tion in dairy goats and supports advancements in the dairy goat industry.] Thank you for pointing this out. We agree with this comment. Therefore, we have rewritten this section. Changes have been made in the manuscript, see lines 651-656. |
Comments 20: [While the conclusion mentions the role of ADCY5 in seasonal reproductive processes, it doesn’t explicitly address the study's initial hypothesis or objectives. Restating the hypothesis and explaining how the results confirm or refute it would create a stronger sense of closure.] |
Response 20: [This study presents evidence that, despite inducing estrus by simulat-ing the light conditions of the breeding season during the non-breeding season, superior follicular de-velopment is observed during the breeding season. The ADCY5 gene emerges as a poten-tial target influencing seasonal follicular development through its regulation of GCs biological functions (Fig.8). This research enhances the understanding of seasonal reproduc-tion in dairy goats and supports advancements in the dairy goat industry.] Thank you for pointing this out. We agree with this comment. Therefore, we have rewritten this section. Changes have been made in the manuscript, see lines 651-656. |
Comments 21: [Figure 7 appears out of place. I doubt if it is suitable for the conclusion or any other section of this paper. It may be better used for literature review manuscripts.] |
Response 21: [It will be deleted or changed] Thank you for pointing this out. We agree with this comment. Therefore, we will contact the journal’s editor, and if it is not appropriate, we will delete or revise it. |
Round 2
Reviewer 1 Report
Comments and Suggestions for Authors
The gene names should be written in italics, but proteins and enzymes should not be in italics. Please check and make corrections throughout the manuscript.
Reviewer 2 Report
Comments and Suggestions for Authors
No comments